**Data Availability Statement:** For metagenomics, all raw reads were submitted to the short read archive under PRJNA564639. The genome of a novel parvovirus was submitted to GenBank (accession MN166196). The genome of a novel

# Viruses in unexplained encephalitis cases in American black bears (*Ursus americanus*)

**Charles E. Alex**[1]©, **Elizabeth Fahsbender**[2,3]©, **Eda Altan**[2,3], **Robert Bildfell**[4,5], **Peregrine Wolff**[6], **Ling Jin**[4,5], **Wendy Black**[5], **Kenneth Jackson**[1], **Leslie Woods**[7], **Brandon Munk**[8], **Tiffany Tse**[1], **Eric Delwart**[2,3], **Patricia A. Pesavento**[1]*

1 Department of Pathology, Microbiology, and Immunology, University of California—Davis School of Veterinary Medicine, Davis, California, United States of America, 2 Vitalant Research Institute, San Francisco, California, United States of America, 3 Department of Laboratory Medicine, University of California—San Francisco, San Francisco, California, United States of America, 4 Department of Biomedical Sciences, Carlson College of Veterinary Medicine, Oregon State University, Corvallis, Oregon, United States of America, 5 Oregon Veterinary Diagnostic Laboratory, Carlson College of Veterinary Medicine, Oregon State University, Corvallis, Oregon, United States of America, 6 Nevada Department of Wildlife, Reno, Nevada, United States of America, 7 California Animal Health and Food Safety Laboratory, Davis, California, United States of America, 8 California Department of Fish and Wildlife, Rancho Cordova, California, United States of America

© These authors contributed equally to this work.
* papesavento@ucdavis.edu

## Abstract

Viral infections were investigated in American black bears (*Ursus americanus*) from Nevada and northern California with and without idiopathic encephalitis. Metagenomics analyses of tissue pools revealed novel viruses in the genera *Circoviridae*, *Parvoviridae*, *Anelloviridae*, *Polyomaviridae*, and *Papillomaviridae*. The circovirus and parvovirus were of particular interest due to their potential importance as pathogens. We characterized the genomes of these viruses and subsequently screened bears by PCR to determine their prevalence. The circovirus (*Ursus americanus* circovirus, UaCV) was detected at a high prevalence (10/16, 67%), and the chaphamaparvovirus (*Ursus americanus* parvovirus, UaPV) was found in a single bear. We showed that UaCV is present in liver, spleen/lymph node, and brain tissue of selected cases by *in situ* hybridization (ISH) and PCR. Infections were detected in cases of idiopathic encephalitis and in cases without inflammatory brain lesions. Infection status was not clearly correlated with disease, and the significance of these infections remains unclear. Given the known pathogenicity of a closely related mammalian circovirus, and the complex manifestations of circovirus-associated diseases, we suggest that UaCV warrants further study as a possible cause or contributor to disease in American black bears.

## Introduction

American black bears (*Ursus americanus*) are the most common and widely distributed bear species in North America. Categorized as a species of "Least Concern" by the International Union for Conservation of Nature, their total population in North America is estimated at approximately 850,000–950,000 and growing [1]. Their adaptability to suburban and urban environments—and the expansion of those environments—brings them into relatively

circovirus was submitted to GenBank (accession MN371255).

**Funding:** The authors received no specific funding for this work.

**Competing interests:** The authors have declared that no competing interests exist.

frequent contact and conflict with human and domestic animal populations. A clear understanding of the diseases affecting black bears is critical to the management of these populations, including mitigation of outbreaks and potential pathogen spillovers. Black bears may be susceptible to various pathogens infecting domestic animals, including several important viruses such as Canine distemper virus, Canine adenovirus 1, and Canine parvovirus. Serology-based surveillance has established that these exposures are not infrequent [2–6]. However, efforts to correlate infection status with clinical disease have been much more limited. Sporadic case reports have described encephalitis in ursid species due to viral etiologies including Canine adenovirus 1, Canine distemper virus, or cross-species transmission of Equine herpesvirus 1 and 9 [7–10]. Cases of clinical disease from Rabies infection appear to be exceedingly rare in bears. In general, disease threats in this species are not well characterized, and the full range of viruses infecting black bears has yet to be thoroughly explored.

A series of cases of idiopathic encephalitis was observed in American black bears from Nevada (2014–2019). Lesions were suggestive of viral etiologies, but diagnostic efforts ruled out known etiologic agents based on histopathology and molecular techniques. Because viral disease was still considered most likely based on histopathologic lesions, we investigated potential viral etiologies by metagenomics analyses. We identified novel viruses in the families *Circoviridae*, *Parvoviridae*, *Anelloviridae*, *Polyomaviridae*, and *Papillomaviridae*. Of these, the circovirus and parvovirus were considered plausible causes of the observed neurologic lesions based on patterns of disease recognized in other species. Encephalitis and cerebellar vasculitis have been associated with Porcine Circovirus infections in pigs, and the human Parvovirus B19 has been associated with a spectrum of neurologic inflammatory lesions in human patients [11–13]. Anelloviruses are highly prevalent, but their association with disease has not been well-established and they are commonly considered to be commensal, asymptomatic infections. Viruses in the *Polyomaviridae* and *Papillomaviridae* families were considered unlikely causes of the observed lesions, and were not pursued. Given their potential pathogenicity, the novel circovirus and parvovirus were further investigated for prevalence by PCR. The novel circovirus was detected at high prevalence, and was further evaluated for tissue distribution and possible lesion association by *in situ* hybridization (ISH).

## Materials and methods

### Animals

The investigation included 17 yearling to adult black bears. Cases 1–8 were free-ranging animals from the Reno and Lake Tahoe areas of Nevada. Case 1 was euthanized because severe clinical disease was observed, and cases 2–7 were euthanized due to behavioral problems (human-bear conflict). Case 8 was killed by a car. For these cases, postmortem examinations took place at the Nevada Department of Agriculture, Animal Disease Lab in Sparks, Nevada. Collected tissues were submitted to the Oregon Veterinary Diagnostic Laboratory in Corvallis, Oregon as part of an investigation of a cluster of cases of non-suppurative encephalitis. That investigation identified a novel gammaherpesvirus, but a specific cause for neurologic disease was not identified [14].

Cases 9–12 were free-ranging bears either found dead or euthanized due to severe disease or human-wildlife conflict from El Dorado, Mendocino, and Santa Barbara counties in California. These cases were necropsied at the California Department of Fish and Wildlife's Wildlife Investigations Laboratory, and tissues were submitted to the California Animal Health and Food Safety laboratory in Davis, CA for histopathology and ancillary testing.

Cases 13–16 had been housed in a wildlife sanctuary in northern California for 10–15 years prior to their deaths. Cases 13–15 were geriatric animals euthanized for progressive mobility

difficulties associated with degenerative joint disease, and case 16 died spontaneously with non-suppurative encephalitis. These cases were necropsied at the UC Davis School of Veterinary Medicine. Case information and significant postmortem findings are summarized in Table 1.

## Metagenomic analyses

Frozen tissues were mechanically homogenized with a handheld rotor in 1 mL of PBS buffer and the homogenate was centrifuged at 9,000 *rpm* for 5 min. Supernatant (500 μl) was placed in a microcentrifuge tube with 100 μl of zirconia beads and quickly frozen on dry ice, thawed, and vortexed five times, then centrifuged for 5 minutes at 9,000 *rpm*. Tissue samples were pooled according to animal and not by tissue type. The supernatants were then passed through a 0.45 μm filter (Millipore, Burlington, MA, USA) and digested for 1.5 hours at 37˚C with a mixture of nuclease enzymes consisting of 14U of Turbo DNase (Ambion, Life Technologies, USA), 3U of Baseline-ZERO (Epicentre, USA), 30U of Benzonase (Novagen, Germany) and 30U of RNase One (Promega, USA) in DNase buffer (Ambion, Life Technologies, USA) to enrich for viral particles. Nucleic acids were extracted immediately afterwards using the Mag-MAX™ Viral RNA Isolation kit (Applied Biosystems, Life Technologies, USA) according to the manufacturer's instructions. Nucleic acids were incubated for 2 min with 100 pmol of random primer A (5′GTTTCCCACTGGANNNNNNNN3′) followed by a reverse transcription step using Superscript III (Invitrogen) with a subsequent Klenow DNA polymerase step (New England Biolabs). cDNA was then further amplified by a PCR step using AmpliTaq Gold™ (ThermoFisher Scientific) DNA polymerase LD with primer B (similar to primer A but minus the randomized 3' end, or 5′GTTTCCCACTGGATA3′). The reaction (25 μL) contained with 2 μM of primer B, 1.85U of AmpliTaq Gold® DNA Polymerase (Applied Biosystems, Thermo Fisher, Waltham, MA, USA), 0.25 of mM dNTPs, 4 mM of MgCl2, 1× PCR Buffer, and 5 μL of cDNA template. The thermal profile for amplification was composed of 95˚C for 5 min, 5

**Table 1. American black bear cases included in this investigation.**

| Case | Location (county) | Sex | Age | Significant post-mortem findings | Euthanized | Date of death or euthanasia |
|------|-------------------|-----|-----|----------------------------------|------------|------------------------------|
| 1 | Washoe, NV | M | 1y | Encephalitis | Y | 30 Jan 2014 |
| 2 | Washoe, NV | F | 1y | Encephalitis | Y | 17 Mar 2014 |
| 3 | CA, near Reno, NV# | M | 1y | Encephalitis | Y | 7 Apr 2017 |
| 4 | Washoe, NV | F | 1y | Encephalitis, tooth root abscess | Y | 21 Feb 2018 |
| 5 | Douglas, NV | M | 3y | Encephalitis | Y | 9 Jul 2014 |
| 6 | Reno/Tahoe, NV# | M | 1y | Mild dermatitis | Y | 24 Apr 2017 |
| 7 | Douglas, NV | F | 1y | Encephalitis | Y | 2 Jul 2018 |
| 8 | Reno/Tahoe, NV# | M | 1y | Trauma (hit by car) | N | 22 Aug 2018 |
| 9 | Mendocino, CA | M | adult | Meningeal lymphoma | Y | 8 Mar 2018 |
| 10 | El Dorado, CA | M | 11m | Hepatic necrosis, encephalitis | N | 25 Dec 2018 |
| 11 | El Dorado, CA | M | 11m | Hepatitis, encephalitis (Sarcocystis) | N | 15 Dec 2018 |
| 12 | Santa Barbara, CA | M | adult | Nasal tumor | Y | 20 Aug 2017 |
| 13 | Calaveras, CA* | F | adult | Degenerative joint disease, nasal tumor | N | 26 Dec 2017 |
| 14 | Calaveras, CA* | M | adult | Degenerative joint disease | Y | 9 Jul 2018 |
| 15 | Calaveras, CA* | M | adult | Degenerative joint disease | Y | 31 Jul 2017 |
| 16 | Calaveras, CA* | F | adult | Encephalitis | Y | 20 Jan 2011 |

*Sanctuary-housed.

#Specific county of origin unknown.

cycles of 95˚C for 30 s, 59˚C for 60 s, and 72˚C for 90 s, 35 cycles of 95˚C for 30 s, 59˚C for 30 s, and 72˚C for 90 s (+2 s per cycle) followed by 72˚C for 10 min and hold at 4˚C. The amplified product was checked by gel electrophoresis (approximately expected size 300–1000 bp), and one ng was used as target for Illumina library generation. The randomly amplified DNA products were quantified by Quant-iT™ DNA HS Assay Kit (Invitrogen, USA) using Qubit fluorometer (Invitrogen, USA). The library was generated using the transposon-based Nextera™ XT Sample Preparation Kit (Illumina, San Diego, CA, USA) and the concentration of DNA libraries was measured by Quant-iT™ DNA HS Assay Kit. The libraries were pooled at equal concentration and size-selected for a range of 300–1,000 bp using the Pippin Prep (Sage Science, Beverly, MA, USA). The library was quantified using the KAPA library quantification kit for Illumina platforms (Kapa Biosystems, USA) and a 10 pM concentration was loaded on the MiSeq sequencing platform for 2x250 cycles pair-end sequencing with dual barcoding. Human and bacterial reads were identified and removed by comparing the raw reads with human reference genome hg38 and bacterial genomes release 66 (collected from ftp://ftp.ncbi. nlm.nih.gov/blast/db/FASTA/, Oct. 20, 2017) using local search mode. The filtered sequences were de-duplicated if base positions 5 to 55 were identical. One random copy of duplicates was kept. The sequences were then trimmed for quality and adaptor and primer sequences by using VecScreen [15]. After that, the reads were de novo assembled by EnsembleAssembler [16]. Assembled contigs and all singlet reads were aligned to an in-house viral protein database (collected from ftp://ftp.ncbi.nih.gov/refseq/release/viral/, Oct. 20, 2017) using BLASTx (version 2.2.7) with E-value cutoff of 0.01. The significant viral hits were then aligned to an in-house non-virus-non-redundant (NVNR) universal proteome database using DIAMOND [17] to eliminate false positive viral hits. Hits with more significant E-value to NVNR than to viral database were removed. Remaining singlets and contigs were compared to all eukaryotic viral protein sequences in GenBank's non-redundant database using BLASTx. The genome coverage of the target viruses was further analyzed by Geneious R11.1.4 software (Biomatters, New Zealand). Genome features were identified by visualization of expected protein sequences of canonical domains, and splice sites were predicted based on expected RNA transcripts.

## PCR detection

For UaCV DNA detection in fresh tissue, DNA was extracted from tissue samples that were collected at necropsies and stored frozen (-80˚C) until use, using the DNEasy Blood and Tissue Kit (Qiagen) according to the manufacturer's instructions. A nested PCR ("Assay 1") was used, consisting of two separate PCR reactions wherein the PCR product from the first round was used as template for the second reaction. The assay targeted part of the Rep gene, based on the contig discovered through metagenomic analysis, with the first-round primers S6_circo_322F (5′-GGCGGGGATTCAAGTGCTAT-3′) and S6_circo_651R (5′-TGGGTTCCCACA GGTAAAGC-3′) to amplify a 329-nt first round product, and second round primers S6_circo_356FN (5′-GGGGTAATTGGTGGGATGGG-3′) and S6_circo_625RN (5′-GCCCTTCAT CCCAGGTAAGG-3′) to amplify a 269-nt second-round product. The PCR [containing a final concentration of 0.2 μm of each primer, 0.2 mM of dNTPs, 0.625 U of Amplitaq Gold® DNA polymerase (Applied Biosystems, Waltham, MA, USA), 1× PCR Gold buffer II, 1.5 mM of MgCl$_2$ and 1 μL of DNA template in a 25 μl reaction] proceeded as follows: 95˚C for 5 min, 40 cycles of (95˚C for 30 s, (52˚C for the first round and 54˚C for the second round of primers) for 30 s, and 72˚C for 30 s), followed by a final extension at 72˚C for 7 min. PCR products of the correct size were verified by gel electrophoresis and Sanger sequencing.

    For formalin-fixed, paraffin-embedded (FFPE) tissues, a separate, single-amplification assay ("Assay 2") was used to amplify a 276-nt segment of the Rep gene using primers

BearCV.92F (5′–CTGACCTTGAAGATGCCTGTAG–3′) and BearCV.368R (5′–CATACC–CATCCCACCAATTACC–3′). This one-step assay amplifies a slightly smaller product than the first round of Assay 1, and was utilized to circumvent problems of viral genomic degradation due to formalin fixation. Multiple serial sections (scrolls) or single 5-µm-thick unstained sections scraped from slides of FFPE liver tissue were deparaffinized in xylene and graded alcohol washes, and DNA was extracted using the DNEasy Blood and Tissue Kit (Qiagen) according to manufacturer instructions. PCR targeting a housekeeping gene (GAPDH) was used to confirm successful DNA extractions. For UaCV "Assay 2," reactions consisted of HotStarTaq Plus Master Mix (Qiagen), 5 pmol of each primer, 2.5 µL of dye, and approximately 25–50 ng of template DNA, diluted to a final volume of 25 µL. Cycling conditions were: 95˚C for 5 minutes, followed by 40 cycles of 94˚C (30s), 52˚C (30s), 72˚C (30s), and a final 72˚C elongation step for 10 minutes. For visualization, products were run on 1.4% agarose gels containing Gel Red (Biotium).

### *In situ* hybridization

To demonstrate viral genome in tissue sections, colorimetric *in situ* hybridization (ISH) was performed on 5-µm-thick sections of formalin-fixed, paraffin-embedded tissues on Superfrost Plus slides (Fisher Scientific, Pittsburgh, PA) using the RNAscope 2.5 Red assay kit (Cat #322360, Advanced Cell Diagnostics, Inc., Hayward, CA). We designed V-UaCirV, (ACD Cat #555001) as 26 ZZ-paired probe sets targeting a 1409 nt segment of the viral genome corresponding to nucleotide positions 478–1887 of the reference sequence (GenBank accession MN371255). Selected tissues included livers and spleens, as these are established sites of circoviral distribution in related species, as well as brains in order to evaluate the possible association with neurologic disease in these cases. Each 5-µm-thick tissue section was pretreated with heat and protease prior to probe hybridization for 2 hours at 40˚C. Negative controls used for validation of signal included an unrelated (GC-content matched) probe on serial sections. Slides were counterstained with hematoxylin and mounted with EcoMount (Biocare Medical, Concord, CA).

## Results

### Metagenomics

Metagenomics analysis was performed on tissues from seven black bears (cases 1–6, 9). The analyzed tissues and results are summarized in Table 2. Metagenomic analysis revealed the presence of a novel circovirus in all but one pool, a polyomavirus in two pools, and a novel papillomavirus and a novel parvovirus each present in one pool. Anellovirus reads were detected in all sample pools. The 812 bp polyomavirus contig in case 3 showed 95% aa identity to the large T antigen of the giant panda polyomavirus (NC_035181.1), while the polyomavirus in case 9 had 74% identity to the *Leptonychotes wedellii* (Weddell seal) polyomavirus large T antigen (NC_032120). The 887 bp papillomavirus contig present in case 9 had 55% identity to L2 *Canis familliaris* papillomavirus (NC 013237). All raw reads were submitted to the short read archive under PRJNA564639.

### New parvovirus

A novel parvovirus, *Ursus americanus* Parvovirus (UaPV, accession MN166196) in the new genus *Chaphamaparvovirus*, was identified in the kidney and liver of case 1. The nearly complete genome is 3,787 bp containing three major open reading frames (ORFs), including a 660 aa non-structural protein (NS1), a 165 aa NP, and the 490 aa viral capsid (VP) (Fig 1). The NP

**Table 2. Viruses detected by metagenomics analyses.**

| Case | Tissue pools | Virus family | Reads per million |
|---|---|---|---|
| 1 | Kidney | *Anelloviridae* | 17950 |
| | | *Circoviridae* | 149 |
| | | *Parvoviridae* | 169 |
| 2 | Cerebrum | *Anelloviridae* | 28 |
| | Lymph node | *Circoviridae* | 3 |
| 3 | Cerebrum | *Anelloviridae* | 774 |
| | | *Circoviridae* | 39 |
| 4 | Cerebrum | *Anelloviridae* | 507 |
| | | *Circoviridae* | 29 |
| 5 | Cerebrum | *Anelloviridae* | 3432 |
| | | *Circoviridae* | 4 |
| 6 | Cerebrum | *Anelloviridae* | 243 |
| | Lymph node | *Polyomaviridae* | 12 |
| 9 | Cerebrum | *Anelloviridae* | 20827 |
| | | *Circoviridae* | 2 |
| | | *Papillomavirus* | 27 |
| | | *Polyomaviridae* | 1 |

was detected in two potential isoforms, including NS2-L (165 aa) and after splicing NS2-P (490 aa). BLASTp of these proteins revealed 73% identity in NS1, 76% identity in VP, 85% identity in NS2-L, and 76% identity in NS2-P to the recently described mouse kidney parvovirus (MKPV) that induces kidney disease in laboratory mice (MH670587) [18]. UaPV is therefore currently the closest relative of MKPV. The *in silico* predicted splicing is similar to that of MKPV based on the putative promotors, predicted splice sites, and polyadenylation signals. There is a conserved acceptor site just before the start of the VP ORF. Two polyadenylation sites were identified based on the conserved nature of these sites in other chaphamaparvoviruses. There are four putative promoter sites reported in this genome, however, the promoter located upstream of the VP start codon is conserved among chaphamaparvoviruses [19]. Putative exons include the NS1, two isoforms of the NS2, and VP.

## New circovirus

The *Ursus americanus* circovirus (UaCV) genome (GenBank accession MN371255) is 2,054 bp with two ambisense open reading frames encoding capsid and replication-associated proteins (Fig 2). Based on full nucleotide sequence, it is most similar to a circovirus identified in a masked palm civet (*Paguma larvata*; Pl-CV8, GenBank accession LC416890.1), with which it shares 75.42% nt sequence identity. A BLASTp showed the Cap (260 aa) had 80% identity to Pl-CV8, and the Rep (287 aa) had a 72% identity to another masked palm civet circovirus sequence, Pl-CV9 (GenBank accession LC416391.1). UaCV contains a stem-loop motif between the intergenic region of the two ORFs consisting of a palindromic 15 bp stem, an 11 bp loop for the initiation of rolling-circle replication, and a 9 bp canonical nonamer (5′-TACTATTAC-3′) on the apex of the loop. The Rep contains 3 rolling circle replication motifs at the N-terminus including, motif I [CFTVNN], motif II [PHLQG], and motif III [YCKK]. The superfamily 3 helicase motifs located at the C-terminus of the UaCV replication-associated protein displayed a Walker-A motif [GPPGCGKT], a Walker–B motif [CLDD], and motif C [ITSN]. Similar to the PlCV, two introns were identified in the

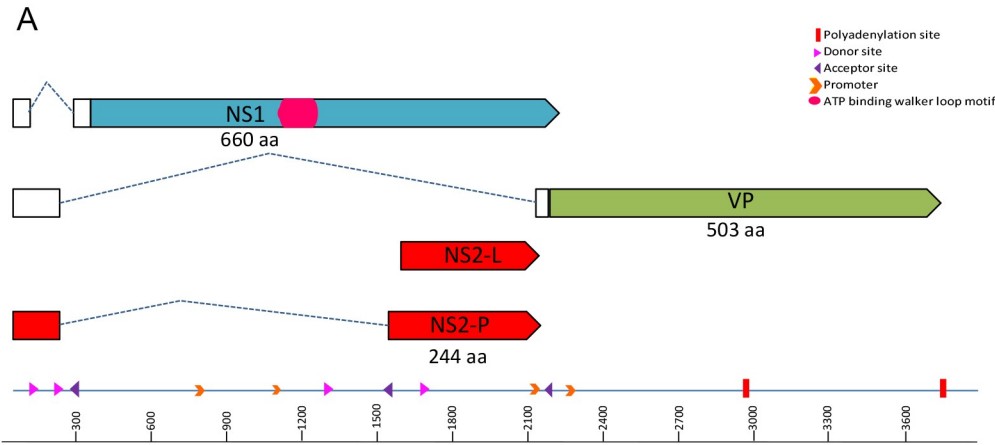

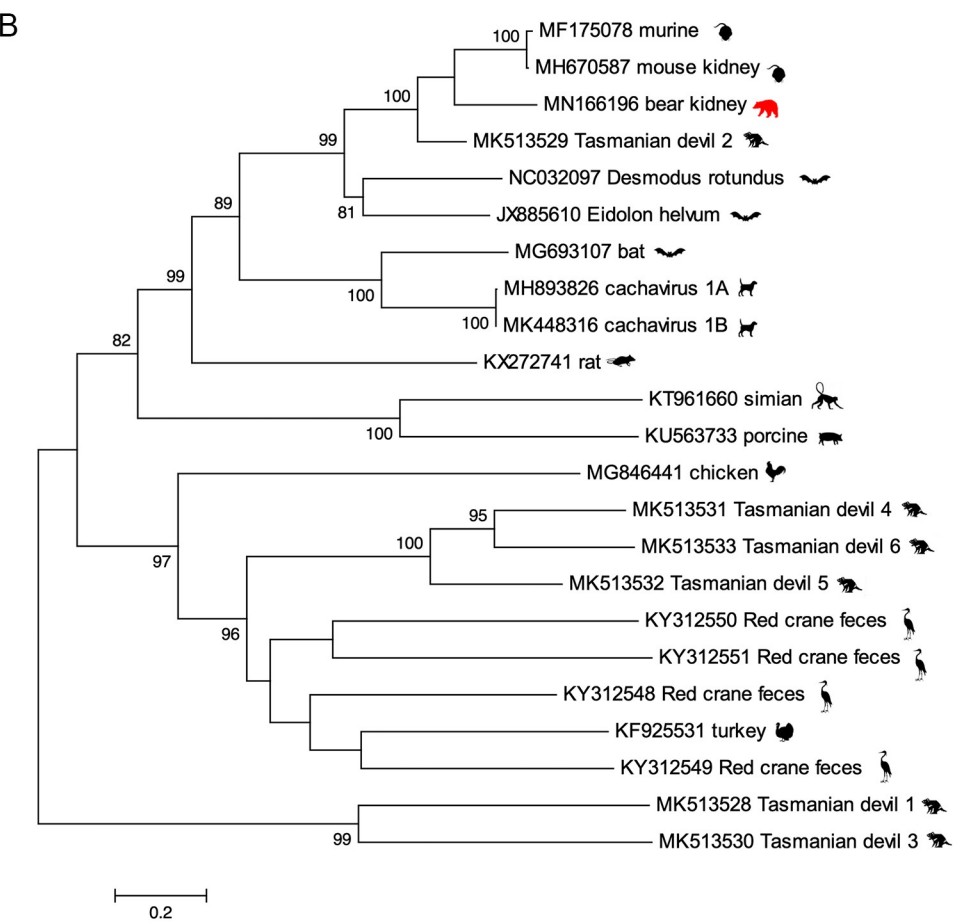

**Fig 1.** A. The genome schematic of UaPV. Clear boxes represent untranslated regions, while the dashed lines represent splicing. The arrows represent ORFs. B. Maximum likelihood tree of NS1 aa chaphamaparvovirus sequences. Bar, 0.2 amino acid substitutions per site. Bootstrap values below 60 were removed.

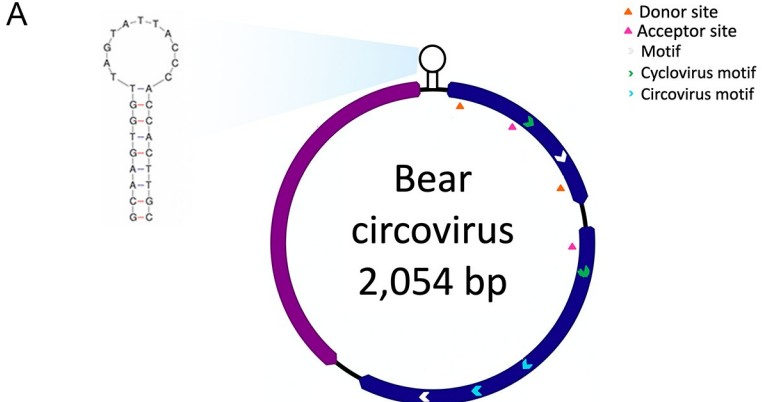

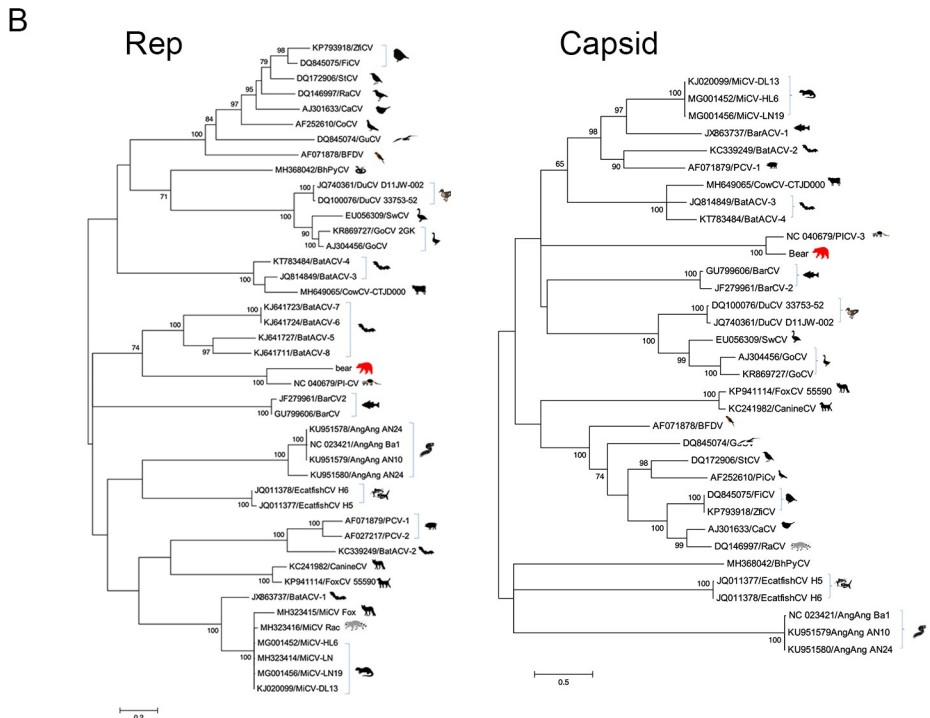

**Fig 2.** A. Genome map of *Ursus americanus* circovirus (UaCV) including Rep (blue) and Capsid (purple) genes. B. Maximum likelihood trees of the Rep (left) and Capsid (right) aa circovirus sequences. Bar, 0.2 (Rep) and 0.5 (Capsid) amino acid substitutions per site. Bootstrap values below 60 were removed.

Rep encoding ORF at locations 47–96 and 363–505. A BLASTx of the Rep showed a 235 aa region that has a 51% identity to porcine circovirus 3 (PCV3-CN-AHB004-2018, accession MK178285.1).

## Prevalence

Having detected UaPV in one case and UaCV in multiple cases by metagenomics, we elected to screen additional black bears for these viruses by PCR. In total, tissues from 16 black bears were tested by PCR for circovirus and parvovirus. For cases 1–6, the same fresh tissue samples used for metagenomics were screened. For cases 7–16, only formalin-fixed, paraffin-embedded

(FFPE) tissues were available. Parvovirus was detected in one case (index case 1). Circovirus was detected in 10/16 cases (62.5%) by one or both PCR assays. Results of PCR detection are summarized in Table 3.

## Circovirus ISH

Given the high frequency of UaCV detection, we evaluated possible tissue targets of infection by *in situ* hybridization. Selected tissues from 12 cases were examined for circovirus by ISH. Positive probe hybridization was detected in 6 cases. Tissues examined included liver, spleen, and brain sections (5 cases), brain only (2 cases), liver only (2 cases), liver and spleen (2 cases), or lymph node only (1 case). Individuals and specific tissues tested were limited by availability of tissue blocks with adequate tissue preservation. Sections examined and results are summarized in Table 4.

Livers that were positive by ISH (4/10) exhibited punctate to diffuse cytoplasmic hybridization in individual cells in hepatic sinusoids (presumed Kupffer cells and/or endothelial cells). This pattern was seen diffusely throughout liver sections (Fig 3), and is consistent with patterns of circovirus detection in the livers of PCV-infected pigs [20] and in dogs infected with Canine circovirus (Pesavento lab, unpublished data).

In 5/9 cases, lymphoid tissues (spleen or lymph node) exhibited similar but more sparse punctate hybridization in individual cells. In spleens, signal was distributed throughout the tissue in the cytoplasm of scattered individual round cells (presumed lymphocytes and/or macrophages). Lymph nodes exhibited a similar pattern, with signal appearing most prominently in cells in the medullary sinuses and occasionally in cortices.

**Table 3. PCR detection of circovirus and parvovirus in black bear tissue samples.**

| Case | Encephalitis | Tissue | Fresh/FFPE | Circovirus PCR | | Parvovirus PCR |
|------|--------------|--------|------------|----------------|-------|----------------|
| | | | | Result | Assay | |
| 1 | Y | Kidney | Fresh | + | 1 | + |
| | | Liver | Fresh | + | 1 | + |
| 2 | Y | Lymph node | Fresh | + | 1 | - |
| | | Cerebrum | Fresh | - | 1 | - |
| 3 | Y | Lymph node | Fresh | - | 1 | - |
| | | Cerebrum | Fresh | - | 1 | - |
| | | Liver | FFPE | - | 2 | - |
| 4 | Y | Cerebrum | Fresh | + | 1 | - |
| | | Liver | FFPE | - | 2 | - |
| 5 | Y | Cerebrum | Fresh | + | 1 | - |
| | | Liver | FFPE | - | 2 | - |
| 6 | N | Cerebrum | Fresh | + | 1 | - |
| 7 | Y | Liver | FFPE | + | 2 | - |
| 8 | N | Liver | FFPE | - | 2 | - |
| 9 | N | Liver | FFPE | + | 2 | - |
| 10 | Y | Liver | FFPE | - | 2 | - |
| 11 | Y | Liver | FFPE | + | 2 | - |
| 12 | N | Liver | FFPE | - | 2 | - |
| 13 | N | Liver | FFPE | + | 2 | - |
| 14 | N | Liver | FFPE | + | 2 | - |
| 15 | N | Liver | FFPE | - | 2 | - |
| 16 | N | Liver | FFPE | - | 2 | - |

**Table 4. Summary of circovirus *in situ* hybridization testing and results.**

| Case | PCR result | PCR-positive tissue | *In situ* hybridization | | |
| --- | --- | --- | --- | --- | --- |
| | | | Liver | Spleen/Lymphoid[a] | Brain |
| 1 | + | Kidney, liver | + | + | +* |
| 2 | + | Lymph node | - | - | -* |
| 3 | - | | - | - | -* |
| 4 | + | Cerebrum | + | + | +* |
| 5 | + | Cerebrum | + | + | NT* |
| 7 | + | Liver | + | + | -* |
| 8 | - | | - | - | + |
| 9 | + | Liver | NT | NT | - |
| 10 | - | | - | NT | NT* |
| 11 | + | Liver | - | NT | NT* |
| 12 | - | | NT | + (LN) | NT |
| 13 | + | Liver | - | - | - |

Positive UaCV ISH probe hybridization was detected in various tissues from a subset of cases. Positive results are highlighted in green.

NT = not tested due to lack of available tissue blocks or lack of adequate tissue preservation.

*Histologic diagnosis of encephalitis.

[a]Spleen tissue was tested whenever possible. In case 12, spleen tissue was unavailable but positive ISH probe hybridization was observed in other lymphoid tissues (lymph nodes).

Brains were of particular interest due to a series of cases of unexplained encephalitis in black bears in the region. Ten of the cases in this study had postmortem diagnoses of mild to severe cerebral cortical inflammation characterized by multifocal to coalescing zones of mononuclear perivascular cuffing, edema, gliosis, hemorrhage, and necrosis. Sections of cerebrum from 8 cases were examined by ISH. Five of these cases had histologic evidence of encephalitis, and three did not. For animals with encephalitis, sections including inflammatory lesions were selected. Sparse positive hybridization was observed in brains from 3 cases—two with encephalitis and one without—occurring as scattered punctate intracellular or extracellular foci. To confirm positive probe hybridization above spurious background staining, two authors (CEA, TT) evaluated ISH slides and negative controls for these sections and tallied individual punctate foci of hybridization, noting their location with respect to blood vessels: vessel-associated (including luminal, endothelial, mural, and Virchow-Robbins space signal) or neuroparenchymal. Counts from the two reviewers were averaged to compare signal from circovirus-specific ISH slides with their corresponding negative controls. Cases were interpreted as positive when the number of puncta of specific probe hybridization far exceeded (>10x) background signal. In all three cases, most hybridization signal was seen in the lumen, endothelium, wall, or Virchow-Robbins spaces of small-caliber blood vessels, suggesting the possibility of endothelial or vascular mural infection. Affected vessels were distributed throughout the examined brain sections, without apparent co-localization to lesions of encephalitis. No clear association between infection and disease was established. No other tissues exhibited similar vessel-associated ISH signal. Counts of ISH signal foci by brain region in these cases are summarized in S1 Table.

## Discussion

A metagenomic investigation of black bears identified novel viruses belonging to five genera: *Circoviridae*, *Parvoviridae*, *Anelloviridae*, *Polyomaviridae*, and *Papillomaviridae*. Of these, circovirus and anellovirus genomes were detected at high prevalence in the study cohort, while

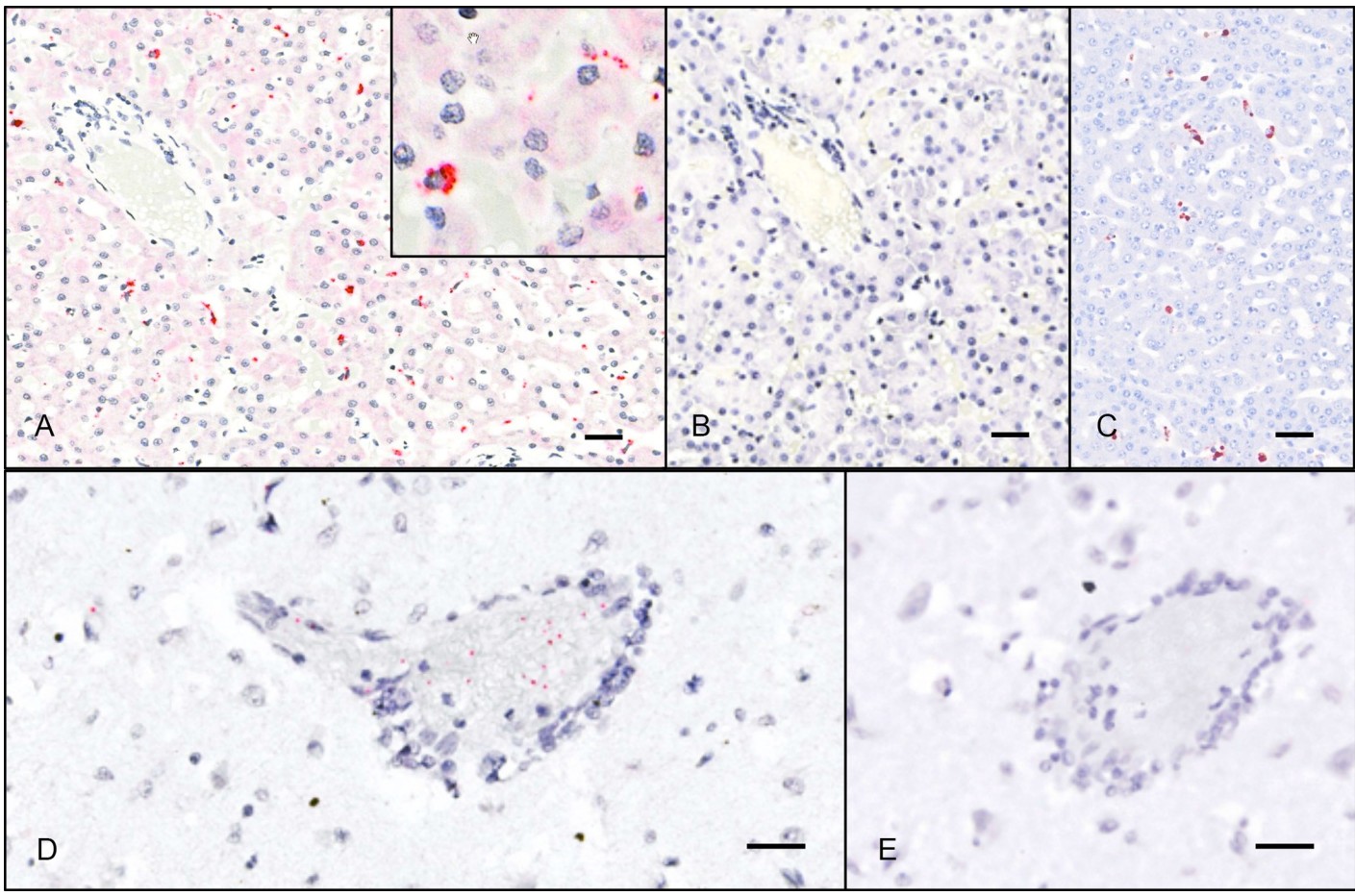

**Fig 3. ISH detection of UaCV nucleic acid in tissue sections.** Case 1. UaCV probes (A, D) and negative control (DapB) probes (B, E); hematoxylin counterstain. A. In sections of liver, UaCV ISH signal is detected in scattered polygonal cells (presumed Kupffer cells or endothelial cells) in hepatic sinusoids. Inset: higher magnification of a cell exhibiting positive probe hybridization. B. No signal is detected with negative control (DapB) probes. C. Immunohistochemistry for PCV2 in the liver of a pig demonstrates a similar pattern of immunoreactivity. (UC Davis SVM archives. IHC performed at California Animal Health and Food Safety Laboratory, Davis CA.) D. Sections of brain exhibit punctate UaCV ISH signal in and around small-caliber blood vessels. E. No signal is detected with negative control probes. Bars = 25μm.

parvovirus, polyomavirus, and papillomavirus genomes were identified in 1–2 cases each. Although a definitive association of viral infection with encephalitis lesions was not established, our results expand the known diversity of viruses infecting black bears, and several of the identified viruses warrant further consideration as potential pathogens.

Circovirus infection was detected in 10/16 (62.5%) of cases tested, but the clinical significance of these infections remains to be established. Within Circoviridae, an 80% nucleotide sequence identity threshold is used for species demarcation [21]. Thus UaCV, which shares 75.42% nt sequence identity with its closest known relative (Pl-CV8), warrants classification as a novel viral species. While both UaCV and Pl-CV8 were identified in carnivore hosts (the latter in a masked palm civet), the two host species are not closely related and do not share a geographic range. There is no evidence to suggest a recent viral spillover from palm civet to black bear; rather, we speculate that UaCV has been enzootic but previously undetected in black bears. Retrospective studies of archived black bear cases from across their geographic range would be useful to clarify the evolutionary history and historical prevalence of UaCV in this population, as well as any association with disease.

The presence and distribution of UaCV sequences in spleens and livers, as demonstrated by PCR and ISH, is consistent with the behavior of known circoviruses in other mammalian hosts, including pathogenic circoviruses of pigs. However, the "classic" histologic lesion of circovirus infection derived from PCV2 studies, characterized by lymphoid depletion with histiocytic replacement and botryoid cytoplasmic inclusions [22], was not evident in these cases. Circoviral nucleic acid was detected by ISH in brains of 3 bears—two with encephalitis and one without—and the distribution in these sections was overwhelmingly vascular or perivascular. It is plausible that the positive ISH signal we observed in these sections indicates a viral tropism for endothelial cells, as has been demonstrated for other circoviruses, but no association could be made between the distribution of ISH signal and histologic brain or vascular lesions. In pigs, PCV2 is known to be endotheliotropic and has been associated with vascular lesions including lymphohistiocytic vasculitis and fibrinoid vascular necrosis, with PCV2 genome demonstrable in endothelium, mural myocytes, and perivascular/infiltrating leukocytes [12]. Although affected vessels may be found anywhere in the body, neurovascular (particularly cerebellar) involvement has been reported in a subset of cases [13]. Similarly, canine circovirus was demonstrated by ISH in (histologically normal) presumed endothelial cells in at least two cases that had evidence of necrotizing vasculitis in other tissues [23]. We speculate that the observed distribution of UaCV in bear brains could be consistent with infection of endothelial cells or vascular mural cells. Alternatively, in encephalitic bears, a compromised blood-brain barrier could allow leakage of virions or virus-infected cells from circulation, accounting for an observed vascular/perivascular distribution. However, the affected blood vessels were often histologically normal by routine staining, and in cases with encephalitis ISH signal was not obviously associated with areas of inflammation.

Several cases exhibited inconsistent circovirus testing results. Disparate results between PCR assays (cases 2, 4, and 5) may be attributable to differences in tissues tested (case 2) and/or degradation of genomic material in formalin-fixed, paraffin-embedded tissues (cases 4 and 5). Several PCR-positive cases were negative by ISH (cases 2, 11, and 13), possibly reflecting low viral load or variation in the tissue distribution of virus. Extended formalin fixation time or advanced autolysis are also plausible causes for loss of ISH signal in tissue sections. Two cases (8 and 12) were PCR-negative for circovirus but exhibited convincing probe hybridization by ISH. In these cases, tissues used for PCR differed from those examined by ISH, and the PCRs also utilized DNA from FFPE tissues, so tissue distribution, low viral concentration, and genomic degradation from prolonged fixation are also possible causes for these results.

The identification of a circovirus infecting black bears expands the known mammalian host range of the *Circoviridae*, and could provide new insight into the disease threats affecting this species. These infections were not definitively associated with pathologic findings. However, given the complicated manifestations of circovirus-associated diseases, we suggest that UaCV warrants further study as a possible cause or contributor to disease in American black bears, particularly (as with other pathogenic mammalian circoviruses) in the context of viral persistence, immunologic effects, and potentiation of/by co-infections. The pathogenic potential of UaCV and other viruses in this study could be influenced by co-pathogens and the general health and immune status of these bears.

Viruses in the genus *Chaphamaparvovirus* are a recent addition to the family *Parvoviridae* [19, 24]. Chaphamaparvovirus infections have been demonstrated to cause chronic kidney disease in laboratory mice, and a chaphamaparvovirus in domestic dogs has been suggested to play a role in diarrhea, but the full spectrum of chaphamaparvovirus-associated disease remains to be elucidated [18, 25]. Species demarcation criteria within *Parvoviridae* dictate that members of a given species share >85% amino acid sequence identity in the nonstructural (NS1) protein [24]. UaPV shares ~73% aa sequence identity with its closest phylogenetic

neighbors (murine chaphamaparvoviruses), and thus warrants classification as a novel parvo-viral species. UaPV was detected in one case in the present study, and its clinical significance is unknown. Given the known association of murine chaphamaparvoviruses with renal disease, further investigations of UaPV focused on renal tissue may be warranted. Sample availability precluded this possibility in the present study.

Anellovirus genomic material was found in all cases investigated by metagenomic analysis. *Anelloviridae* is a ubiquitous family of small, single-stranded, circular DNA viruses that are highly prevalent in the human population (>90%) and cause persistent, presumably life-long infections [26]. Anelloviruses are also ubiquitous in other mammals and generally considered commensal infections [27]. In pigs, Torque teno sus anellovirus viral loads are increased when co-infected with porcine circovirus 2 [28] and anellovirus loads also increase in immunosuppressed humans [29–36]. The possible contribution to neurologic lesions or other disease in these bears remains unknown, and the high rate of detection could reflect immune compromise in some of these cases.

The papillomavirus and the two distinct polyomaviruses identified in this investigation were also found in 1 and 2 cases, respectively, and were considered unlikely to have been associated with significant apparently clinical disease in this cohort.

In summary, we identified diverse novel viruses infecting black bears from California and Nevada. A clear relationship between infection status and neurologic lesions was not established for any of these viruses. However, this work expands the known diversity of viruses infecting free-ranging black bears in north America, and several of the identified viruses warrant further investigation as potential pathogens.

## Supporting information

**S1 Table. Foci of probe hybridization in cases exhibiting ISH signal in sections of brain.** Signal was distributed throughout sections, but predominantly identified in/around small-caliber blood vessels (lumen, endothelium, wall, or Virchow-Robbins space). Counts are averages from two reviewers.
(DOCX)

## Acknowledgments

We thank Vorthon Sawasong for helpful discussions, and Carl Lackey and Drs. Terza Brostoff and Steven Kubiski for reviewing the manuscript. We thank the field and wildlife health staff at CDFW and NDOW for their technical and logistical support in handling these cases.

## Author Contributions

**Conceptualization:** Charles E. Alex, Elizabeth Fahsbender, Eric Delwart, Patricia A. Pesavento.

**Formal analysis:** Charles E. Alex, Elizabeth Fahsbender, Eric Delwart, Patricia A. Pesavento.

**Investigation:** Charles E. Alex, Elizabeth Fahsbender, Eda Altan, Peregrine Wolff, Ling Jin, Wendy Black, Kenneth Jackson, Leslie Woods, Brandon Munk, Tiffany Tse, Eric Delwart, Patricia A. Pesavento.

**Methodology:** Elizabeth Fahsbender, Kenneth Jackson.

**Resources:** Robert Bildfell, Peregrine Wolff, Leslie Woods, Brandon Munk.

**Supervision:** Robert Bildfell, Kenneth Jackson, Eric Delwart, Patricia A. Pesavento.

**Visualization:** Elizabeth Fahsbender.

**Writing – original draft:** Charles E. Alex, Elizabeth Fahsbender.

**Writing – review & editing:** Charles E. Alex, Elizabeth Fahsbender, Robert Bildfell, Eric Delwart, Patricia A. Pesavento.

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
