## [Decision Letter · Decision Letter 0]

9 Jun 2020

PONE-D-20-14538

Viruses in unexplained encephalitis cases in American black bears (*Ursus americanus*)

PLOS ONE

Dear Dr. Pesavento

Thank you for submitting your manuscript to PLOS ONE. After careful consideration, we feel that it has merit but does not fully meet PLOS ONE’s publication criteria as it currently stands. Therefore, we invite you to submit a revised version of the manuscript that addresses the points raised during the review process.

Many thanks for submitting your manuscript to PLOS One

Your manuscript was reviewed by two experts in the field, who have both made some suggestions for improvement prior to acceptance

If you could write a detailed response to reviewers, this will expedite review when it is resubmitted

I wish you all the best with your revisions

Hope you are keeping safe and well in these difficult times

Thanks

Simon

We look forward to receiving your revised manuscript.

Kind regards,

Simon Clegg, PhD

Academic Editor

PLOS ONE

2. We note that you are reporting an analysis of a microarray, next-generation sequencing, or deep sequencing data set. PLOS requires that authors comply with field-specific standards for preparation, recording, and deposition of data in repositories appropriate to their field. Please upload these data to a stable, public repository (such as ArrayExpress, Gene Expression Omnibus (GEO), DNA Data Bank of Japan (DDBJ), NCBI GenBank, NCBI Sequence Read Archive, or EMBL Nucleotide Sequence Database (ENA)). In your revised cover letter, please provide the relevant accession numbers that may be used to access these data. For a full list of recommended repositories, see http://journals.plos.org/plosone/s/data-availability#loc-omics or http://journals.plos.org/plosone/s/data-availability#loc-sequencing.

Reviewers' comments:

Reviewer's Responses to Questions

**Comments to the Author**

1. Is the manuscript technically sound, and do the data support the conclusions?

Reviewer #1: Yes

Reviewer #2: Partly

2. Has the statistical analysis been performed appropriately and rigorously? 

Reviewer #1: N/A

Reviewer #2: N/A

3. Have the authors made all data underlying the findings in their manuscript fully available?

Reviewer #1: Yes

Reviewer #2: Yes

4. Is the manuscript presented in an intelligible fashion and written in standard English?

Reviewer #1: Yes

Reviewer #2: Yes

5. Review Comments to the Author

Reviewer #1: Alex C, et al. describes in the manuscript entitled “Viruses in unexplained encephalitis cases in American black bears Ursus americanus” the identification of diverse viruses in fresh and formalized tissue samples from black bears of Nevada and California states. They have used a metagenomic approach for viral identification, and PCR and ISH for viral frequency determination, tissue distribution and possible pathological associations. Two putative new viruses from Circoviridae and Parvoviridae families have been detected and the nearly complete genome sequence described. Potential genomic structures and putative proteins have been predicted and the viral evolutionary history has been analyzed including representative viruses members.

The manuscript fits the journal scope, it is well structured, the results are clear and the figures are didactic and easy to understand, however, there are some concerns that must be addressed before proceeding.

Questions

1. Do you think that using viral RNA extraction kit in fresh samples may have affected your results, since you have described the detection of DNA viruses.

2. You are proposing the identification and description of two new viruses, the UaPV and UaCV. Can you please include the ICTV demarcation criteria for these two new viral species?

3. Don’t you need an Ethic committee approval for this work?

Minor comments

Material and Methods

Table 1. Can you include the collection date of the samples? It is a good information for time-spatial identification of viral presence;

Line 94. What dilution (weight/volume) were the samples prepared? How tissues were homogenized for tissue disruption? Mechanical, syringe, automated...? Please, include this information in the text;

Line 96. What do you mean by vortexed for five cycles? After that, was it again clarified at 9,000rpm during 5 min? If it was, please include this information in the text;

Line 101. Did you follow the manufacture instructions or and adaptation for RNA isolation? Please, include this information;

Line 103. Please, can you comment why you have used “primer A” for cDNA synthesis instead of generically random primers? Please, also comment why did you use only “primer A-short” for DNA amplification.

Line 131-134. Can you explain why did you used two round PCR for circovirus detection? What are the targets of these primers?

Line 142. Why did you use a different PCR assay (Assay 2) for circovirus detection in formalin-fixed tissue?

Results

Line 204 – 209 and 220 - 227. How did you predict all those features? Splicing sites, polyadenylation signals, promoter sites, ATP biding motif, stem-loop motif, rolling circle motif and all other motifs? Please, comment in Material and Methods.

Figure 2. Can you please include which color represents the Rep and Capsid ORF (Blue/purple)?

Table 4. Which tissue are you considering as Positive/negative for PCR in this table? Since we have different PCR results from different tissues in Table 3, it is not clear which tissue you are referring to. Please, clarify;

Writing suggestions

Line 53. “…We identified novel viruses from Circoviridae, Parvoviridae, Anelloviridae, Polyomaviridae, and Papillomaviridae families….”;

Line 56. Remove the sentence “This report describes the results of these investigations.” ;

Line 62. “… due to behavioral problems (human-bear conflict) or due to severe clinical disease observation (Case 1)…”;

Line 100. Remove “1x”;

Line 105. Please, include manufacturer of AmpliTaq Gold;

Line 112. “…using BLASTx (version 2.2.7) with E-value cutoff…”

Line 123. “The significant viral hits were then…”;

Line 124. “…to eliminate false positive viral hits.”;

Line 130. “For UaCV DNA detection in fresh tissue,…”;

Line 145. Can you change “Scrolls” for a more formal word? In addition, specify what is FFPE, since it is the first time of appearance;

Line 149. Please, include the Celsius degree symbol “°” in all temperatures;

Line 196. First time appearance of UaPV, please describe the whole name;

Line 379. “…increased when co-infected with pig circovirus 2…”;

Line 385. “…associated with significant apparently clinical disease…”;

Reviewer #2: Alex and Fahsbender et al., present a manuscript detailing an investigation of the virome of American black bears resident in Nevada and California using a high through put sequencing approach. Using this technique the authors found novel members of the Circo-, Parvo-, Anello-, Polyoma- and Papillomaviridae in the American black bear. The authors then examined the novel Circovirus and Parvovirus in more detail including experiments to evaluate if the novel Circovirus may contribute to cases of encephalitis with unknown causes. This paper contributes to our understanding of viral pathogens of this species, the data is freely available via NCBI, and the paper is generally well presented and readable.

Points to Address:

While the paper is readable and well presented the structure and language used infers a link between the novel ursine Circovirus and encephalitis that the data does not support and the authors state this conclusion clearly towards the end of the manuscript. For me the presentation of the data needs revision to remove this inferred link except where it is directly addressed experimentally to avoid ambiguity for the reader.

Could the authors comment more on why they chose to focus on the circovirus and parvovirus to the exclusion of the anello/polyoma/papillomaviruses? Circovirus, with sequences detected in many individuals was an obvious candidate to follow up, anellovirus sequences were detected in all tested individuals (often with the highest number of reads) but were not followed up, why? Polyomavirus sequences were detected in two individuals with papillomavirus and parvovirus in 1 case each. With this in mind could the rationale of choosing parvovirus rather than papillomavirus or polyomavirus be made clear? To this end revision of the introduction to present existing data on which this choice may have been based would be helpful, as would indication of if any of these agents have known links to encephalitis. Similarly more detail on known ursine viral pathogens (with or without links to encephalitis) would aid the reader in interpreting the significance of the authors work and choice in further investigating the novel ursine circo- and parvovirus.

The methods section would benefit from revision also:

Within the animals section it may aid the reader if the animals and handling of samples is presented in order ie origin of cases 1-8, then handling/processing, followed by the same for 9-16 to make this section easier to read.

Within the metagenomics section additional detail concerning sample preparation would assist in reproduction and interpretation. Additional detail on how samples were initially homogenised and why a second homogenisation with zircon beads was performed would be helpful. Clarity is also needed on if these are fresh, frozen or FFPE preserved samples, and if fresh how rapidly after isolation treatment occurred and any steps taken to inhibit endogenous nucleases.

Can the authors explain why they incubated their homogenate with a cocktail of 2 DNases, one broad spectrum nuclease and an RNase when seeking to purify viral RNA? Repeated freeze thaws and use of RNases may have decreased the author’s ability to isolate the genomes of novel ursine RNA viruses, and decreased transcript abundance of the DNA viruses identified. Also why isolate only RNA which for DNA viruses would not be protected at all from nucleases by encapsidation?

With all viruses except anelloviruses having low reads/million counts additional controls are necessary to show these are not contamination. Procedural control with blank samples to test for contamination from kits would be important in this context as would those to assess the survival of spiked in known (uncoated) RNAs/DNAs through homogenisation and nuclease treatment to see if these process account for the low read counts of most identified viruses.

Additional detail is needed for the cDNA synthesis, DNA amplification and library preparation stages, for example how much DNA is used, the source etc. How was library cDNA isolated and what concentrations were the pools, what sizes were selected?

For the PCR assays please make clear if the given primers are based on the authors own UaCV sequence or are degenerate based on other known viruses. Are the Nested primers used in a single reaction or in separate reactions, this isn’t clear – and if used in separate reactions how has the template DNA been treated between reaction with primer set 1 and 2? What is the source of tissue DNA in this context and how much is used to template the PCR? When using FFPE samples how much input material is used for DNA extraction?

For the ISH probes can coordinates for the target region be given with reference to the authors’ genomic sequence?

Results:

Please define what is meant by pools (L173/174). Can sequence comparison of the anellovirus data also be given here, and can data for the coverage of genomes be given? In table 2 does reads per million relate to short reads or contigs?

Can the authors make clear that Pl-CV9 is from a masked palm civet as Pl-CV8, and increase the size of the stem loop structure in Fig 2A to make it readable. Would the authors comment on the proximity of their novel ursine Circovirus to a circovirus of masked palm civet, do these animals share a geographical range? Is there brain tissue (as FFPE) available for performing the UaCV PCR on in cases 7-16 to investigate the putative link with encephalitis further?

In table 4 does not tested (NT) mean not available or not done?

For the ISH data where regions of brain were examined were these taken near to or from zones of encephalitis? The probe detects genomic DNA, probes targeting viral mRNAs may be more appropriate here to show active infection. Why is DapB used as a control and with multiplexing possible with this technology can an internal positive ISH control be used (against a host gene) to better assess the validity of the ISH detection of vDNA/viral mRNA. This technique can also be used quantitatively assess the levels of nucleic acid, have the authors considered using this approach particularly on their brain tissue sections to further support their conclusions?

6. PLOS authors have the option to publish the peer review history of their article (what does this mean?). If published, this will include your full peer review and any attached files.

Reviewer #2: No

---

## [Author Response · Author response to Decision Letter 0]

25 Nov 2020

Thank you for the opportunity to revise this manuscript. Our specific responses to reviewers' questions and suggestions are below, and were also included with the submission as a separate Word document (from which the responses below were copied and pasted). 

Reviewer #1: Alex C, et al. describes in the manuscript entitled “Viruses in unexplained encephalitis cases in American black bears Ursus americanus” the identification of diverse viruses in fresh and formalized tissue samples from black bears of Nevada and California states. They have used a metagenomic approach for viral identification, and PCR and ISH for viral frequency determination, tissue distribution and possible pathological associations. Two putative new viruses from Circoviridae and Parvoviridae families have been detected and the nearly complete genome sequence described. Potential genomic structures and putative proteins have been predicted and the viral evolutionary history has been analyzed including representative viruses members.

The manuscript fits the journal scope, it is well structured, the results are clear and the figures are didactic and easy to understand, however, there are some concerns that must be addressed before proceeding.

Questions

1. Do you think that using viral RNA extraction kit in fresh samples may have affected your results, since you have described the detection of DNA viruses.

The kit used (MagMAXTM Viral RNA Isolation kit) is based on both RNA and DNA binding to silica. Both RNA and DNA are purified despite the kit name with the DNA typically being removed by DNAse digestion. From one of the vendor’s web site: 

The MagMAX™ Viral RNA Isolation Kit can efficiently isolate viral RNA and DNA from samples as large as 400 µl. RNA and DNA recovery is typically greater than 75%, but may vary depending on sample type. The RNA recovered with the kit is of high quality and purity and is suitable for real-time RT-PCR.

2. You are proposing the identification and description of two new viruses, the UaPV and UaCV. Can you please include the ICTV demarcation criteria for these two new viral species?

Species demarcation criteria for both viral families are now included in the discussion section of the revised manuscript. 

3. Don’t you need an Ethic committee approval for this work?

An ethics statement has been added to the submission. This retrospective study was performed on stored/archived samples from diagnostic cases at the behest of the California Department of Fish and Wildlife (CDFW). No live animals were used to support this study; all samples were taken post-mortem from dead or dying animals. When euthanasia was performed it was either done under the auspices of the Wildlife Resource Agency for public safety or animal welfare concerns or by an attending veterinarian for animal welfare concerns. For tissues from CDFW cases, all were taken as part of the normal duties performed by the Wildlife Health staff and processed and examined in collaboration with other state agencies, laboratories (e.g. CAHFS), and research institutions (e.g. UCD).

Minor comments

Material and Methods

Table 1. Can you include the collection date of the samples? It is a good information for time-spatial identification of viral presence;

This information has been added to the table.

Line 94. What dilution (weight/volume) were the samples prepared? 

How tissues were homogenized for tissue disruption? Mechanical, syringe, automated...? Please, include this information in the text;

 We have now indicated in the text that homogenization was performed mechanically. 

Line 96. What do you mean by vortexed for five cycles? After that, was it again clarified at 9,000rpm during 5 min? If it was, please include this information in the text;

 Additional detail has been added to Materials and Methods to clarify this process: “Frozen tissues were mechanically homogenized with a handheld rotor in 1 mL of PBS buffer and the homogenate was centrifuged at 9,000 rpm for 5 min. Supernatant (500 µl) was placed in a microcentrifuge tube with 100 µl of zirconia beads and quickly frozen on dry ice, thawed, and vortexed five times, then centrifuged for 5 minutes at 9,000 rpm. The supernatants were then passed through a 0.45 µm filter (Millipore, Burlington, MA, USA) and digested for 1.5 hours at 37°C...”

Line 101. Did you follow the manufacture instructions or and adaptation for RNA isolation? Please, include this information;

Manufacturer suggestions were followed; this has now been indicated in the text.

Line 103. Please, can you comment why you have used “primer A” for cDNA synthesis instead of generically random primers? Please, also comment why did you use only “primer A-short” for DNA amplification.

Primer-A behaves in a very similar fashion to random hexamers due to its randomized 3’ end. “Primer-A short” is the same primer minus the randomized 3 prime and is used to further amplify the products of primer-A. “Primer-A-short” was renamed “Primer B” in the revised manuscript for clarity. 

Line 131-134. Can you explain why did you used two round PCR for circovirus detection? What are the targets of these primers?

The nested approach was utilized for sensitive detection in tissues in which viral genetic material may have been present at low copy number. We have clarified in the manuscript that these primers target the Rep gene.

Line 142. Why did you use a different PCR assay (Assay 2) for circovirus detection in formalin-fixed tissue?

Assay 2 amplifies a smaller first-round product than the first step of Assay 1, so is potentially more sensitive for use in FFPE tissues where viral genomic DNA degradation could have impaired detection. A statement of this rationale was added to the methods sections. 

Results

Line 204 – 209 and 220 - 227. How did you predict all those features? Splicing sites, polyadenylation signals, promoter sites, ATP biding motif, stem-loop motif, rolling circle motif and all other motifs? Please, comment in Material and Methods.

These features were detected based on visualization of the expected sequences and canonical domains. Splice sites were similarly detected based on expected RNA transcripts. A statement to this effect has been added to the Materials and Methods section. 

Figure 2. Can you please include which color represents the Rep and Capsid ORF (Blue/purple)?

 The figure legend was updated to include this information.

Table 4. Which tissue are you considering as Positive/negative for PCR in this table? Since we have different PCR results from different tissues in Table 3, it is not clear which tissue you are referring to. Please, clarify;

 This information has been added to the table.

Writing suggestions

Line 53. “…We identified novel viruses from Circoviridae, Parvoviridae, Anelloviridae, Polyomaviridae, and Papillomaviridae families….”;

Line 56. Remove the sentence “This report describes the results of these investigations.” ;

Line 62. “… due to behavioral problems (human-bear conflict) or due to severe clinical disease observation (Case 1)…”;

Line 100. Remove “1x”;

Line 105. Please, include manufacturer of AmpliTaq Gold;

Line 112. “…using BLASTx (version 2.2.7) with E-value cutoff…”

Line 123. “The significant viral hits were then…”;

Line 124. “…to eliminate false positive viral hits.”;

Line 130. “For UaCV DNA detection in fresh tissue,…”;

Line 145. Can you change “Scrolls” for a more formal word? In addition, specify what is FFPE, since it is the first time of appearance;

Line 149. Please, include the Celsius degree symbol “°” in all temperatures;

Line 196. First time appearance of UaPV, please describe the whole name;

Line 379. “…increased when co-infected with pig circovirus 2…”;

Line 385. “…associated with significant apparently clinical disease…”;

All of these changes have been implemented in the revised manuscript.

Reviewer #2: Alex and Fahsbender et al., present a manuscript detailing an investigation of the virome of American black bears resident in Nevada and California using a high through put sequencing approach. Using this technique the authors found novel members of the Circo-, Parvo-, Anello-, Polyoma- and Papillomaviridae in the American black bear. The authors then examined the novel Circovirus and Parvovirus in more detail including experiments to evaluate if the novel Circovirus may contribute to cases of encephalitis with unknown causes. This paper contributes to our understanding of viral pathogens of this species, the data is freely available via NCBI, and the paper is generally well presented and readable.

Points to Address:

While the paper is readable and well presented the structure and language used infers a link between the novel ursine Circovirus and encephalitis that the data does not support and the authors state this conclusion clearly towards the end of the manuscript. For me the presentation of the data needs revision to remove this inferred link except where it is directly addressed experimentally to avoid ambiguity for the reader.

 We appreciate and share the reviewer’s concern for clarity regarding disease causation. We have tried to be very cautious in asserting any link between viral infection and disease in these cases. We have included statements in the abstract and results of the revised manuscript to further clarify that no definitive disease association has yet been established. 

Could the authors comment more on why they chose to focus on the circovirus and parvovirus to the exclusion of the anello/polyoma/papillomaviruses? Circovirus, with sequences detected in many individuals was an obvious candidate to follow up, anellovirus sequences were detected in all tested individuals (often with the highest number of reads) but were not followed up, why? Polyomavirus sequences were detected in two individuals with papillomavirus and parvovirus in 1 case each. With this in mind could the rationale of choosing parvovirus rather than papillomavirus or polyomavirus be made clear? To this end revision of the introduction to present existing data on which this choice may have been based would be helpful, as would indication of if any of these agents have known links to encephalitis. Similarly more detail on known ursine viral pathogens (with or without links to encephalitis) would aid the reader in interpreting the significance of the authors work and choice in further investigating the novel ursine circo- and parvovirus.

We have included more detail in the introduction regarding our rationale for pursuing the circovirus and parvovirus, and not the others. Additional detail was also included regarding viral pathogens in bears, including those linked to encephalitis. 

The methods section would benefit from revision also:

Within the animals section it may aid the reader if the animals and handling of samples is presented in order ie origin of cases 1-8, then handling/processing, followed by the same for 9-16 to make this section easier to read.

 This section has been re-structured according to this recommendation. 

Within the metagenomics section additional detail concerning sample preparation would assist in reproduction and interpretation. Additional detail on how samples were initially homogenised and why a second homogenisation with zircon beads was performed would be helpful. Clarity is also needed on if these are fresh, frozen or FFPE preserved samples, and if fresh how rapidly after isolation treatment occurred and any steps taken to inhibit endogenous nucleases.

We have added details of the virus enrichment sample preparation. Procedure was used on thawed frozen samples. Here the goal is to enrich viral nucleic acids within viral particles. Both homogenization and the use of zircon beads and repeated freezing and thawing are methods we have found that help release viral particles from cells and tissues and allow then to pass through 0.45ul filter. Endogenous nucleases were not inhibited as the goal is to minimize the concentration of all nucleic acids except those protected from digestion within viral particles. We actually add nuclease enzymes before extracting particle-protected nucleic acids using kits that contain denaturants that will then denature these nucleases.

Can the authors explain why they incubated their homogenate with a cocktail of 2 DNases, one broad spectrum nuclease and an RNase when seeking to purify viral RNA? Repeated freeze thaws and use of RNases may have decreased the author’s ability to isolate the genomes of novel ursine RNA viruses, and decreased transcript abundance of the DNA viruses identified. Also why isolate only RNA which for DNA viruses would not be protected at all from nucleases by encapsidation?

The extraction kit used is based on nucleic acid binding to silica. Both RNA and DNA are purified. The RNA specificity claimed by the kit’s name (MagMax iral RNA isolation kit) requires use of DNAse AFTER extraction which we do not use. We therefore preferentially purify and extract both viral RNA AND DNA. Two DNAses and one RNAse are used to maximize digestion of host and bacterial nucleic acids in the homogenates while viral nucleic acids are protected from digestion within their viral capsids prior to their extraction. The initial use of reverse transcription to generate cDNA ensure that our methodology is able to amplify both RNA and DNA viruses as reported in prior publications. 

With all viruses except anelloviruses having low reads/million counts additional controls are necessary to show these are not contamination. Procedural control with blank samples to test for contamination from kits would be important in this context as would those to assess the survival of spiked in known (uncoated) RNAs/DNAs through homogenisation and nuclease treatment to see if these process account for the low read counts of most identified viruses.

The metagenomics procedure was done to identify viral nucleic acids rather then to quantify them. We acknowledge the possibility that other viruses present at lower concentrations could have been missed. The presence of both the circovirus and the parvovirus were confirmed in new extracts by PCR. 

Additional detail is needed for the cDNA synthesis, DNA amplification and library preparation stages, for example how much DNA is used, the source etc. How was library cDNA isolated and what concentrations were the pools, what sizes were selected?

Because the quantities of nucleic acids remaining after the extensive particle enrichment/purification steps are minimal we are typically unable to measure them by conventional methods and simply proceed with the random amplification as described. One ng of the random RT-PCR amplification step is used as target for the Illumina Nextera step as now added to the materials and methods. 

For the PCR assays please make clear if the given primers are based on the authors own UaCV sequence or are degenerate based on other known viruses. Are the Nested primers used in a single reaction or in separate reactions, this isn’t clear – and if used in separate reactions how has the template DNA been treated between reaction with primer set 1 and 2? What is the source of tissue DNA in this context and how much is used to template the PCR? When using FFPE samples how much input material is used for DNA extraction?

We have clarified in the text that PCR primers were based on sequences generated by metagenomic analysis, and that the nested PCR consisted of two separate, sequential reactions, wherein the product of reaction 1 was used as template for reaction 2. We also clarified that DNA extracted from fresh tissue samples (stored frozen since necropsy) was used for these assays. 

For assays on FFPE, DNA was extracted from tissue scrolls when possible, or from single unstained tissue sections scraped from glass slides. We confirmed successful DNA extraction using PCR targeting a housekeeping gene (GAPDH) and used a spectrophotometer to quantify extracted DNA to normalize the PCR reactions. The PCR reactions used approximately 25-50 ng of template DNA. 

For the ISH probes can coordinates for the target region be given with reference to the authors’ genomic sequence? 

This information was added to the Materials and Methods section. 

Results:

Please define what is meant by pools (L173/174). 

The text has been updated to reflect that tissue samples were pooled by animal, and not by tissue type. 

Can sequence comparison of the anellovirus data also be given here, and can data for the coverage of genomes be given? 

 As a frequent infection generally considered to be a commensal we did not perform further analyses of the anellovirus sequences.

In table 2 does reads per million relate to short reads or contigs?

These numbers refer to short reads. 

Can the authors make clear that Pl-CV9 is from a masked palm civet as Pl-CV8, and increase the size of the stem loop structure in Fig 2A to make it readable. 

We have edited the text to clarify that Pl-CV9 is also from a masked palm civet. The figure has been adjusted as suggested. 

Would the authors comment on the proximity of their novel ursine Circovirus to a circovirus of masked palm civet, do these animals share a geographical range? 

We have included comments on the relationships between UaCV and the palm civet circovirus in the revised manuscript, and have also indicated that there is no overlap in geographic ranges of these two carnivore hosts. The palm civet sequences were detected from animals in Japan. 

Is there brain tissue (as FFPE) available for performing the UaCV PCR on in cases 7-16 to investigate the putative link with encephalitis further?

 Particularly for brain tissue, there were some inconsistencies among cases in terms of autolysis, samples collected, and duration of formalin fixation that could affect PCR results. Given that no obvious link was established – which we have endeavored to make clear in the manuscript – we elected to forego UaCV PCR testing of FFPE brain tissue in favor of testing prospectively collected samples in which sampling and sample handling are more controlled. We anticipate that this will allow for a more rigorous assessment of viral presence/association with lesions. This work is in progress. 

In table 4 does not tested (NT) mean not available or not done?

NT indicates tissues that were either not available for testing (no available tissue blocks) or tissues with significant autolysis that would have impeded ISH detection or interpretation. 

For the ISH data where regions of brain were examined were these taken near to or from zones of encephalitis? The probe detects genomic DNA, probes targeting viral mRNAs may be more appropriate here to show active infection. Why is DapB used as a control and with multiplexing possible with this technology can an internal positive ISH control be used (against a host gene) to better assess the validity of the ISH detection of vDNA/viral mRNA. This technique can also be used quantitatively assess the levels of nucleic acid, have the authors considered using this approach particularly on their brain tissue sections to further support their conclusions?

 In cases of encephalitis, sections of brain including inflammatory lesions were selected for ISH, although the degree of inflammation and extent of affected tissue did vary between section. We use DapB probes as a control to evaluate levels of spurious, non-specific background staining. Our confidence in the determination of positive signal is based on detection of hybridization in the context of biologically (anatomically) sensible distribution, as evaluated by board-certified veterinary pathologists. For this initial characterization, we feel confident in our interpretation of circoviral presence in these tissues using these methods. We agree that viral mRNA probes, double-labeling, and quantitative methods would provide further support for the presence of active circovirus infection. Additional prospective circovirus studies are planned to further evaluate the possible link between circovirus and encephalitis lesions in black bears, and some of these techniques may be incorporated in that work to support any observed link.

---

## [Editor Report · Decision Letter 1]

3 Dec 2020

Viruses in unexplained encephalitis cases in American black bears (*Ursus americanus*)

PONE-D-20-14538R1

Dear Dr. Pesavento,

We’re pleased to inform you that your manuscript has been judged scientifically suitable for publication and will be formally accepted for publication once it meets all outstanding technical requirements.

Kind regards,

Simon Clegg, PhD

Academic Editor

PLOS ONE

Additional Editor Comments:

Many thanks for resubmitting your manuscript to PLOS One

As you have addressed all the comments, and the manuscript reads well, I have recommended the manuscript for publication

You should hear from the Editorial office soon

It was a pleasure working with you, and I wish you the best of luck for your future research

Hope you are keeping safe and well in these difficult times

Thanks

Simon

---

## [Editor Report · Acceptance letter]

7 Dec 2020

PONE-D-20-14538R1 

Viruses in unexplained encephalitis cases in American black bears (*Ursus americanus*) 

Dear Dr. Pesavento:

I'm pleased to inform you that your manuscript has been deemed suitable for publication in PLOS ONE. Congratulations! Your manuscript is now with our production department. 

Kind regards, 

on behalf of

Dr. Simon Clegg 

Academic Editor

PLOS ONE